# Phenotypic Biomarkers of Aqueous Extracellular Vesicles from Retinoblastoma Eyes

**DOI:** 10.3390/ijms252111660

**Published:** 2024-10-30

**Authors:** Anne Amacker, Chen-Ching Peng, Nan Jiang, Shreya Sirivolu, Nikki Higa, Kevin Stachelek, Bibiana Reiser, Peter Kuhn, David Cobrinik, Paolo Neviani, Jesse L. Berry, Tijana Jovanovic-Talisman, Liya Xu

**Affiliations:** 1The Vision Center, Children’s Hospital Los Angeles, Los Angeles, CA 90027, USA; aamacker@usc.edu (A.A.); ppeng@chla.usc.edu (C.-C.P.); sirivolu@usc.edu (S.S.); kstachelek@chla.usc.edu (K.S.); breiser@usc.edu (B.R.); dcobrinik@chla.usc.edu (D.C.); jesse.berrymd@gmail.com (J.L.B.); 2USC Roski Eye Institute, Keck School of Medicine, University of Southern California, Los Angeles, CA 90033, USA; 3Department of Cancer Biology and Molecular Medicine, Beckman Research Institute, City of Hope Comprehensive Cancer Center, Duarte, CA 91010, USA; najiang@coh.org (N.J.); ttalisman@coh.org (T.J.-T.); 4Michelson Center for Convergent Bioscience, Convergent Science Institute in Cancer, University of Southern California, Los Angeles, CA 90089, USA; nhiga@usc.edu (N.H.); pkuhn@usc.edu (P.K.); 5The Saban Research Institute, Children’s Hospital Los Angeles, Los Angeles, CA 90027, USA; 6Norris Comprehensive Cancer Center, Keck School of Medicine, University of Southern California, Los Angeles, CA 90033, USA; 7The Extracellular Vesicle Core, Children’s Hospital Los Angeles, Los Angeles, CA 90027, USA; pneviani@chla.usc.edu

**Keywords:** retinoblastoma, extracellular vesicles, aqueous humor, CD133, molecular phenotyping

## Abstract

Recent advancements in aqueous humor (AH) cell-free DNA (cfDNA) genomics have opened new avenues for ex vivo molecular profiling of retinoblastoma (RB), the most common pediatric intraocular malignancy, where biopsy is typically prohibited. While these insights offer a genetic blueprint of the tumor, they lack multi-omic molecular phenotyping, which is essential for understanding the functional state. Extracellular vesicles (EVs), naturally present in AH, are promising by offering time-resolved phenotypic information. We employed multiplex bead-based flow cytometry and Single Extracellular Vesicle Nanoscopy (SEVEN) to analyze EV phenotypes in AH from a cohort of five RB, with three uveal melanoma (UM) and two age-matched glaucoma (GLC) samples serving as controls. The studies identified CD133-enriched EVs uniquely in RB AH, absent in both GLC and UM AH. This was corroborated by further analysis of five RB cell lines, including two commercial (Y79, Weri) and three in-house developed lines, confirming CD133 enrichment and supporting its role as an RB-specific EV marker. Single-vesicle analysis demonstrated a strong association of CD133 with CD81 and CD63, with minimal CD9 presence. These results, validated through complementary techniques, position CD133 as a critical marker in RB-derived EVs, paving the way for enhanced multi-omic RB characterization and potential advancements in clinical diagnostics.

## 1. Introduction

Retinoblastoma (RB), the most common pediatric intraocular malignancy [1], typically presents in early childhood. While rare, RB accounts for 3% of all childhood cancers [2]. It can manifest in one eye (60%) or both (40%), resulting in approximately 300 cases diagnosed annually in the United States and 8000 worldwide [3,4]. The primary treatment objective is to save the child’s life and, when safe, preserve the child’s eye with some degree of vision. However, a significant clinical challenge arises due to the lack of effective molecular biomarkers for diagnosis and prognosis. RB tumor biopsy is contraindicated in clinical practice due to the inherent risks associated with extraocular cancer spread, resulting in a profound void in our in vivo understanding of this disease [5,6,7,8,9,10,11]. Although RB was the first cancer to have its genetic etiology identified [12], RB patients have yet to benefit from precision medicine approaches due to the absence of in vivo data. This has led to uncertainties in RB treatment outcomes, underscoring the clinical need for innovative methodologies. In 2017, our laboratory made a significant breakthrough by identifying aqueous humor (AH), an ocular fluid, as a promising source of tumor-derived cell-free DNA (cfDNA). We and other groups have demonstrated that there is sufficient tumor DNA in the cfDNA from AH, with advancements showing recurring somatic copy number alterations (SCNAs), matches in chromosomal gains and losses between tumors and AH, and detection of pathogenic variants in the *RB1* gene [13]. AH has also been compared to blood samples, demonstrating a higher yield of cfDNA and the presence of tumor-derived SCNAs that were lacking from blood [13]. However, there are limitations to analysis due to factors such as decreasing DNA concentrations with treatment, lack of detectable SCNAs in some RB eyes, uncertainty if small tumors may yield enough cfDNA for analysis, and lack of phenotypic features [13,14,15,16]. This was demonstrated in Gerrish et al., as AH samples from the patients treated with intravitreous injection had lower concentrations of cfDNA and could not be accurately quantified (<0.100 ng/μL) [16].

The identification of extracellular vesicles (EVs) as pivotal mediators of intercellular communication and potential biomarkers has opened new avenues of exploration [17,18]. These critical mediators are found in various body fluids, including the AH [19], and play pivotal roles in cellular communication, immunity, and physiological regulation [20]. Their lipid bilayer membrane renders them suitable as a biomarker [20]. Particularly, small EVs (sEVs) have been demonstrated to play a significant role in tumor formation, progression, and treatment resistance [21,22], contributing to cancer-specific diagnostic and prognostic markers across different cancer types [22,23,24,25,26]. Tumor-derived sEVs have been shown to participate in the creation of a niche environment by carrying donor-specific proteins and miRNAs necessary for tight cell junction formation and angiogenesis [27]. On the reverse, sEVs can modify endothelial cells and destroy tight junctions to promote vascular leakiness and alter the extracellular matrix to promote new vasculature. They also induce cells to differentiate into cancer-associated fibroblasts that, in turn, can create EVs for cancer migration and invasion and induce the proliferation of epithelial-mesenchymal transition cells that are aggressive with stem cell-like properties [27]. In ocular cancers, tumor-derived exosomes within the trabecular meshwork have been found to influence signaling pathways, such as those associated with epithelial to mesenchymal transition, angiogenesis, and metastasis [28]. Ongoing research explores the potential of sEVs as retinoblastoma (RB) biomarkers, supported by proteomic profiles of exosomes released from primary RB tumors and vitreous seeding cell lines that have been found to exhibit upregulated proteins associated with factors such as extracellular matrix remodeling, invasion, metastasis, and chemoresistance [29]. However, their utility within AH has received limited explorations [18,20,30]. EVs can provide insights into tumor phenotypes by capturing features of the RB tumor microenvironment—a realm previously beyond the reach of genomics alone.

Integral Membrane protein markers like tetraspanins (e.g., CD9, CD63, and CD81) are common EV biomarkers due to their widespread presence and functional importance [31]. These are expressed in many types of EVs across many body fluids, not just in cancer. Recent studies from our group with single particle interferometric reflectance imaging sensor (SP-IRIS) demonstrated that the AH harbors sEVs, most notably mono-CD63+ sEVs, which may be an AH-specific biomarker [32]. Subsequently, we found that CD63/81+ sEVs were abundant in pre-treatment RB eyes, particularly those with higher tumor burden, suggesting CD63/81+ sEVs are tumor-derived, showcasing their potential as biomarkers [19]. However, while tetraspanins serve as useful pan-EV surface markers, they may not provide the precision needed to discern RB-derived EVs (RB-EVs) from others—or, more importantly, to distinguish disease subtypes within RB. Hence, the use of a highly multiplexed surface marker panel that includes cancer biomarkers is essential for the accurate identification of RB-derived EV subtypes. This study aims to explore specific tumor-sEV membrane surface biomarkers in RB AH samples, providing time-resolved phenotypic information about the disease. Such insights may advance precision management and personalized medicine in future RB treatment.

## 2. Results

### 2.1. Surface Marker Profiling of EVs from AH Samples

Two congenital glaucoma (GLC) AH samples, serving as negative controls for RB, three uveal melanoma (UM) AH samples, and five RB AH samples taken at the time of primary enucleation (PE) without any medical intervention were analyzed. RB patient demographics and clinical characteristics are summarized in Table 1.

MACSPlex, a multiplex bead-based flow cytometry assay, was used to analyze the expression profiles of surface markers found on EVs in AH samples. With 39 capture beads coated with different monoclonal antibodies for 37 different EV surface antigens and two internal negative controls, the fluorescence intensity for each marker could be measured by flow cytometry (Figure 1A). To test the feasibility of using unprocessed aqueous humor in the MACSPlex experiment, we compared the raw AH sample to the supernatant after 10,000× *g* centrifugation. The results showed similar fluorescent profiles obtained from both samples (Appendix A).

All ten unprocessed samples were analyzed through the MACSPlex-flow cytometry protocol. MACSPlex analysis of the 2 GLC AH samples demonstrated a strong CD63 dominance (Figure 1B and Figure 2). When comparing the enrichment ratios of CD63 to the mean tetraspanin MFI, GLC AH samples had a ratio of 2.6 (SD = 0.34) versus RB AH samples mean ratio of 1.3 (SD = 0.27) (*p* = 0.095) (Figure 2). In comparison to the GLC patients, the analysis of EVs in the AH of the 5 RB samples exhibited a much larger co-dominance of CD63/CD81 positivity (Figure 1D and Appendix A), consistent with past research on RB EVs [19]. The mean CD63 MFI for the 5 samples was 11,747, and the mean CD81 MFI was 14,202.6. The mean CD81 enrichment ratio for the five RB samples was 1.5 (SD = 0.26) vs. 0.3 (SD = 0.24) for the two GLC samples (*p* = 0.095) (Figure 2A). Interestingly, 4 out of the 5 RB AH samples demonstrated very strong CD133 signals (Figure 2A). For these four CD133+ samples, the mean CD133 MFI was 58,072.5. This CD133+ was not reflected in the Uveal Melanoma aqueous humor samples used for comparison (Figure 1C and Figure 2). The CD133 ratio compared to the mean tetraspanin MFI for the 3 UM AH samples was 0.04 (SD = 0.04), compared to 5.4 (SD = 5.05) for the 5 RB AH samples (*p* = 0.161) (Figure 2).

### 2.2. Surface Marker Profiling of EVs from RB Cell Lines

To further investigate the relationship between surface markers and RB, five RB cell lines were analyzed using MACSPlex-flow cytometry, including commercial cell lines Y79 and WERI-RB1 and early passage CHLA cell lines VCRB_20, VCRB_24, VCRB_46. MACSPlex analysis of EVs in the RB cell line supernatants again showed a higher occurrence of CD63/81 codominance. The mean CD63 MFI was 10,727.4, with an enrichment ratio of 1.2 (SD = 0.37) (Figure 2A). The mean CD81 MFI was 14,688.6, with an enrichment ratio of 1.5 (SD = 0.19) (Figure 2A). All 5 RB cell lines demonstrated CD133 positivity, with a mean enrichment ratio of 4.3 (SD = 3.02), which was significantly different than compared with GLC and UM combined (*p* = 0.008). While all RB cell lines included in the analysis demonstrate enriched CD133 signal, there is heterogeneity among samples (Figure 2B). The CD133 ratio compared to mean tetraspanin MFI for all 5 samples ranges from 1.0 in VCRB_46 to 8.5 in VCRB_24 (Figure 2B).

### 2.3. CD133 Association with CD63/81 in RB AH EVs

Due to sample scarcity, we focused on Case 79 AH to further explore the association of known EV tetraspanins (CD9, CD63, and CD81) with the CD133 signal in RB samples. MACSPlex protocol with single APC-conjugated antibodies revealed that CD133+ vesicles also demonstrated a high signal with CD81 and CD63 but a low signal with CD9 (Figure 3A). Single-channel MACSPlex experiments revealed the EV subpopulations based on the positivity of CD9, CD63, and CD81 (Figure 3C). A dominant CD63/81 subpopulation was detected RB AH (Figure 3C, 42.8%).

These findings were compared to the analysis of a single particle-interferometric reflection imaging sensor (SP-IRIS). SP-IRIS technology allows the quantification of various populations of EVs in a sample by using fluorescently tagged antibodies against surface marker tetraspanins. Representative fluorescent images detected by fluorescent-conjugated antibodies (Figure 3B) demonstrate increased red and green fluorescent signals, representing CD63 and CD81, respectively, compared to a blue signal which represents CD9. The EV subpopulation percentages can be calculated accordingly using single and double-positive EV populations (Figure 3C) and single, double, and triple-positive EV populations (Appendix A). Concordance between MACSPlex and Exoview results (Figure 3C) affirm a dominant CD63/81 EV subpopulation in RB AH across different platforms. As SP-IRIS evaluates single vesicles, the results suggest that CD63 or CD81 is co-expressed with CD133 on single vesicles.

Nanoparticle Tracking Analysis (NTA) revealed that most particles in AH, including EVs, from Case 79 were <200 nm (Appendix A), typical for extracellular vesicles.

### 2.4. Multiparametric Characterization of RB AH EVs

Next, we used SEVEN to provide a multiparametric characterization of EVs. Representative images in Figure 4A reveal well-defined EVs with very little background fluorescence. In agreement with other data, CD63/CD81-enriched EVs were significantly more abundant compared to CD9-enriched EVs (Figure 4B). Interestingly, we detected a significantly higher number of CD133-enriched EVs when anti-CD133 Ab was used in addition to anti-TSPAN Abs for staining (Figure 4B), suggesting high molecular CD133 content on individual EVs. As expected, in control experiments, we detected a negligible number of EVs on the anti-IgG surface (anti-TSPAN stain) and on the anti-CD63/CD81 surface (no stain). All detected EVs had relatively small sizes (below 260 nm) with low coefficient of variation (between 24 and 26%); details are provided in Appendix A. CD9-enriched EVs were the largest with an average diameter of 80 nm, while CD133-enriched EVs were the smallest with an average diameter of 73 nm (with detection of TSPAN; anti-TSPAN stain) and 74 nm (with detection of TSPAN and CD133; anti-TPAN and anti-CD133 stain). Detected molecule counts per EV were largely uniform; CD133-enriched EV stained with anti-TSPAN Abs that had somewhat lower values. All EVs were largely circular, although CD63/CD81- and CD9- enriched EVs (compared to CD133-enriched EVs) had slightly higher circularity values.

## 3. Discussion

CD63/81+ sEVs were notably abundant in pre-treatment RB eyes, especially those with higher tumor burden, indicating their potential as tumor-derived biomarkers [19]. However, with tetraspanin only, it’s impossible to distinguish RB-derived EVs (RB-EVs) from others and identify specific disease subtypes within RB. This emphasizes the need for a highly multiplexed surface marker panel incorporating cancer-specific biomarkers. Utilizing an optimized multiplexed magnetic bead-based flow cytometry assay, we observed a dominant CD63 APC fluorescence profile in two glaucoma (GLC) samples. Cataract AH has also been investigated in MACSplex, illustrating a CD63 dominance with no prominent CD133 signal [33]. In contrast, all AH samples from RB patients exhibited CD63/81 dominance compared to CD9 enrichment levels, consistent with our previous SP-IRIS research, further validating the outcomes obtained through the new MACSPlex workflow.

While tetraspanin frequencies were vital for evaluating the efficacy of the new workflow, the primary aim of implementing MACSPlex was to explore biomarkers beyond the previously investigated tetraspanins. Remarkably, four out of the five included RB AH samples exhibited significant CD133 fluorescence. Another crucial step in our analysis was the comparison of RB AH samples to other ocular cancers. In our experiment, we included three UM AH samples taken prior to any treatment. In every UM AH sample included, there was no prominent CD133 signal. MACSplex analysis conducted on early-stage, mid-stage, and late-stage retinal pigment epithelium sEVs showed minimal CD133+ as well [34].

Further analysis using MACSplex protocol on samples taken from RB cell lines demonstrated that all five included cell lines were CD133+, further validating the connection between RB EVs and CD133. This is not the first time that commercial cell lines such as Y79 [35] and Weri RB1 [36] have been investigated in relation to CD133 signals. In Nair et al., study results suggested that the low CD133 cell surface signal population within the Rb Y79 cell line was correlated to specific characteristics such as size, colony-forming ability, invasiveness potential, drug resistance, and gene expression pattern. Interestingly, they also found that serial passaging of low CD133 colonies revealed increased levels of CD133 expression with time. Figure 2B demonstrated heterogeneity of expression levels of CD133 between the EVs from 5 RB cell lines examined, with Rb Y79 EVs showing a lower CD133 enrichment ratio than Weri RB EVs. These differences in enrichment ratios may suggest differences in clinical factors for these lines, such as treatment resistance, but surely require a very sophisticated and fit-in purpose study to investigate. However, it is important to note that all cell lines included in these studies were taken at the time of primary enucleation. Therefore, conducting new studies with time-resolved sampling over the course of systemic treatment in non-enucleated eyes would be crucial to further explore this hypothesis.

The patient who tested negative for this marker, Case 34, had several distinct clinical features. Firstly, Case 34 displayed markedly lower intraocular pressure (IOP) than other patients in this cohort (Table 1). Additionally, the absence of tumor seeding into the vitreous chamber suggests a potentially lower tumor burden or less active tumor dissemination, which might result in fewer tumor-derived markers in the aqueous humor. These factors may have contributed to the absence of CD133-specific EVs in this case. However, other factors may have influenced this patient’s primary enucleation, including a tumor size exceeding 15 mm. Further investigations exploring the correlation between CD133 expression and clinical factors in a larger cohort have the potential to enrich our understanding of retinoblastoma evolution and the clinical utility of this marker.

To further investigate the association between CD133 and retinoblastoma, we conducted additional experiments using an ample AH sample from CD133-positive Case 79. Initial experiments with a fluorescent antibody cocktail (CD9, CD63, and CD81) provided EV specificity. Subsequent single-channel experiments demonstrated a strong association between CD133 and CD81, followed by CD63, while CD9 showed a very low level of association with CD133. These findings underscore the potential of CD81 and CD63 as key biological markers for ocular malignancy, with CD133 adding specificity in the context of Retinoblastoma. Comparisons with SP-IRIS and Exoview experiments affirmed the validity of the MACSPlex methodology. It is important to note that Exoview and MACSPlex all rely on the presence of CD9, CD63, or CD81. With this limitation, the potential existence of tetraspanin-negative CD133-EVs must be acknowledged.

This is the first study to use MACSPlex on RB eyes to identify the phenotypic surface markers in AH EV research. Although preliminary, the potential of EVs providing direct access to genomic and phenotypic data from RB tumors is clear. EVs’ role in cancer diagnosis [37] and therapeutic response [38] has been explored in a multitude of cancers. CD133, a pentaspan transmembrane protein, has been established as a marker for cancer stem cells in solid tumors [39]. This study opens promising avenues for future research, such as establishing diagnostic CD133 screening for RB. Further potential applications include minimal residual disease monitoring, as the natural abundance of EVs in the aqueous humor allows us to monitor changes, including the clearance of tumor-derived EVs and the restoration of CD63 dominance. Further validation of this initial data using MACSPlex technology on a larger cohort will deepen our understanding of this disease-associated marker. This case study represents a stride in the evolution of RB management and illustrates the emergence of innovative techniques and discoveries that have the potential to enhance patient care across a spectrum of diseases.

## 4. Materials and Methods

### 4.1. Patients and Sample Collection

This study was conducted under the IRB approval at USC and Children’s Hospital Los Angeles (CHLA). CHLA maintains the following active CHLA IRB approval: CHLA-17-00248, Retinoblastoma Patient Clinical Database, and Tissue Biorepository, which was last re-approved on 31 January 2024. This project adheres to all standards of legal, ethical, and regulatory requirements in the use of human biospecimens for research. The approved protocol ensures that research activities are conducted in compliance with the U.S. Department of Health and Human Services regulations (45 CFR Part 46) and the U.S. Food and Drug Administration (21 CFR Parts 50 and 56). The project complies with the Health Insurance Portability and Accountability Act (HIPAA) and other applicable federal and state regulations governing the protection and use of human subjects, data, and biospecimens in research.

All patients included in the analysis provided written informed consent for the biorepository at CHLA via a legal guardian. AH data was kept separate from clinical data until the final retrospective analysis. AH, samples were collected via clear corneal paracentesis from two congenital glaucoma eyes at diagnosis, three uveal melanoma (UM) samples, and five RB eyes immediately after enucleation. The congenital glaucoma eyes serve as a non-cancerous age-matched control, while the UM samples serve as an intraocular malignancy comparison for RB samples. All AH samples have been collected and stored at −80 °C for analysis.

### 4.2. Cell Culture

CHLA-VC-RB20, CHLA-VC-RB24, and CHLA-VC-RB46 cell lines were established as described in Stachelek et al., 2023 and propagated in 96-well dishes with 200 μL RB culture medium ((IMDM (Mediatech, Manassas, VA, USA)) 10% FBS, 2 mM glutamine (Mediatech, Manassas, VA, USA)), 55 μΜ beta-mercaptoethanol (Sigma–Aldrich, St. Louis, MO, USA), and 10 μg/mL insulin (Humulin-R, Lilly, Indianapolis, IN, USA)) at 37 °C and 5% CO_2_/95% room air [40,41]. Approximately two-thirds of the media changed approximately every two days when the media approached the amber color. To optimize growth, undissociated cell clusters were kept at densities sufficient to acidify the media to around pH 7.0 and were passaged as needed to prevent acidifying beyond pH 6.5 by splitting by no more than 2:3. Y79 cells were cultured as described by Nair et al., 2017 [35]. Weri RB1 cells were cultured as previously described by Hu et al., 2012 [36]. Two centrifugation spins were conducted to collect supernatants and isolate cell supernatant EVs. First, the samples were spun at 2000× *g* for 10 min, and then 10,000× *g* for 10 min. We do not have Institutional Review Board approval to incubate the enucleated eyes from the patient samples included in this study into cell lines due to regulatory limitations.

### 4.3. Broad-Based Multiplex Flow Cytometry Assay (MACSPlex)

AH and cell supernatant samples were subjected to bead-based multiplex analysis by flow cytometry (MACSPlex Exosome Kit, human, Miltenyi, Gaithersburg, MD, USA) per the manufacturer’s instruction. Briefly, 5 to 20 µL of AH or cell supernatant was added to a 1.5 mL tube. 100–125 µL of MACSPlex Buffer Solution was added to each tube depending on whether it was used for a single-channel or triple-channel analysis to acquire a total volume consistent the following day. MACSPlex Exosome Capture Beads (CD1c, CD2, CD3, CD4, CD8, CD9, CD11c, CD14, CD19, CD20, CD24, CD25, CD29, CD31, CD40, CD41b, CD42a, CD44, CD45, CD49e, CD56, CD62P, CD63, CD69, CD81, CD86, CD105, CD133, CD142, CD146, CD209, CD326, HLA-ABC, HLA-DRDPDQ, MCSP, ROR1, and SSEA-4) were vortexed for 30 s and 15 µL were transferred to each tube. All tubes were then incubated overnight at room temperature and protected from the light using an orbital shaker at 450 rpm. The following day, 500 µL of MACSPlex Buffer was added to each tube and then centrifuged at room temperature for 3000× *g* for 5 min. 500 µL of supernatant was carefully aspirated. For single channel analyses, 5 µL of MACSPlex Exosome Detection Reagent CD9, CD63, or CD81 was added to the respective tube. For triple channel analyses, 15 µL of mixed CD9, CD63, and CD81 cocktails were added and mixed by pipetting up and down. The samples were then incubated for 1 h using the same protocol above. Following incubation, 500 µL of MACSPlex Buffer was added to each tube, and tubes were centrifuged at room temperature at 3000× *g* for 5 min. 500 µL of supernatant was then aspirated, leaving 150 µL in the tubes. 500 µL of buffer was once again added, and the tubes were incubated for 15 min using the same protocol. Tubes were then centrifuged at room temperature for 3000× *g* for 5 min, and 500 µL of supernatant was aspirated. 350 µL of MACSPlex Buffer was added to each tube to bring the total volume up to 500 µL, and the samples were transferred into flow cytometry tubes. Flow cytometry analysis was performed using the FACSymphony S6 cell sorter (Becton Dickinson, Franklin Lakes, NJ, USA) available at the CHLA TSRI Flow Cytometry Core.

The median fluorescence intensity (MFI) values of each marker were corrected for background signal by subtracting the respective MFI values from the non-EV-containing buffer samples included in every session of analysis (Figure 1 and Figure 3). For the five RB AH samples included, these MFI plots were later scaled to 150,000 MFI for visualization (Appendix A). Finally, the MFI values were normalized to the mean value of the CD9/CD63/CD81 tetraspanins’ MFI to determine the relative levels of each marker in Figure 2. Additional experiments were later conducted to investigate outliers and mean MFIs for all 39 antigens (Appendix A).

### 4.4. Single Particle-Interferometric Reflectance Imaging Sensor (SP-IRIS) Analysis

SP-IRIS analysis was performed with the ExoView Human Tetraspanin Kit (Unchained Labs, Pleasanton, CA, USA) following the manufacturer’s instructions. This workflow for AH has been described in our previous publications [32]. A desired volume of sample was obtained using 0.02 µL of unprocessed AH diluted in buffer A to a final volume of 40 µL. Then, 35 µL of the sample was incubated on an ExoView Tetraspanin Chip for 16 h. Chips were then washed and incubated with immunocapture antibodies (anti-CD9 CF488, anti-CD81 CF555, and anti-CD63 CF647) based on the manufacturer’s protocol (Unchained Labs, Pleasanton, CA, USA). Chips were imaged with the ExoView R100 reader using ExoView Scanner version 3.2 acquisition software; data were analyzed with the ExoView Analyzer version 3.2.

### 4.5. Nanoparticle Tracking Analysis

This workflow has previously been described in past publications from this laboratory [32]. The nanoparticle size and concentration from 10 µL unprocessed AH samples were diluted in PBS and analyzed using the Nanoparticle tracking analysis system NanoSight NS300 (The Malvern Panalytical, Malvern, UK) equipped with a 405 nm laser and a sCMOS camera. Brownian movement of the particles in suspension was recorded by the camera, and the moving tracks were analyzed using the Stokes-Einstein equation to obtain the hydrodynamic radius and vesicle count for each modal size. Results were displayed as particle count per size distribution. Particle concentration was then calculated based on the input volume. Data analysis was performed using NTA software 3.4, which presented the average and standard deviation of at least five acquisitions.

### 4.6. Photophysical Characterization of Fluorescently Labeled Antibodies

The following antibodies: anti-CD9 (BioLegend, 312102, San Diego, CA, USA), anti-CD63 (Novus Biologicals, NBP2-42225, Centennial, CO, USA), anti-CD81 (BioLegend, 349502, San Diego, CA, USA), anti-CD133 (Miltenyi-Biotec 130-108-062, Gaithersburg, MD, USA ), were labeled with Alexa Fluor 647 dye (AF647; Invitrogen, A20006, Waltham, MA, USA) as described, with degree of labeling ranging between 1 and 2 [42]. The average number of localizations generated per fluorescent probe (alpha) was calculated as described previously [43,44]. Mixture of labeled anti-CD9, -CD63, and -CD81 antibodies had alpha of 14 [43]; mixture of labeled anti-CD9, -CD63, -CD81, and -CD133 antibodies had alpha of 11 (Appendix A).

### 4.7. Single Extracellular Vesicle Nanoscopy (SEVEN) Imaging and Analysis

EVs from the AH sample were assessed with a Single Extracellular Vesicle Nanoscopy (SEVEN) assay as described before [43]. For antibody spotting, 0.5 µL of antibody solution containing 1% glycerol was pipetted onto the center of the MCP4-coated coverslips. We used one of the following: (1) the mixture of anti-CD63 and anti-CD81 antibodies (each at 0.5 mg/mL), (2) anti-CD9 antibody (0.5 mg/mL), (3) anti-CD133 antibody (1 mg/mL), or (4) control goat anti-rabbit IgG antibody (0.5 mg/mL; Invitrogen, A16112, Waltham, MA, USA). Antibodies were incubated for 4 h in a humidity-controlled dish at room temperature, then blocked and washed as described before [43].

AH was diluted in PBS containing 0.025% (*v*/*v*) Tween-20 in the following ratios: (1) 1 to 600 for anti-rabbit IgG spot; (2) 1 to 6000 for anti-CD9 spot; (3) 1 to 6000 for anti-CD63/CD81 spot; (4) 1 to 1200 for anti-CD133 spot. 60 µL of diluted sample was used for experiments. After overnight incubation and wash, captured EVs were stained with antibodies labeled with AF647 in 2% (*w*/*v*) BSA and 0.025% (*v*/*v*) Tween-20 in PBS: either with a mixture of 10 nM anti-CD9, 10 nM anti-CD63, and 10 nM anti-CD81 (anti-TSPAN) or a mixture of 10 nM anti-CD9, 10 nM anti-CD63, 10 nM anti-CD81, and 10 nM anti-CD133. After staining and washing, EVs were fixed as before [43]. The coverslips were loaded into Attofluor cell chambers (Thermo Fisher Scientific, A7816, Waltham, MA, USA) and imaged using N-STORM super-resolution microscope (Nikon Instruments, Melville, NY, USA, components described before [43]). A total of 25,000 frames with 10 ms of exposure time were acquired using NIS-Elements software (Nikon Instruments, Melville, NY, USA), with 41 × 41 μm (256 × 256 pixels) regions of interest (ROIs). Image processing was performed with the N-STORM Offline Analysis Module of the NIS-Elements software (version 5.41.0) as before [43]. The data were analyzed with Matlab R2023a (MathWorks; Natick, MA, USA) as before [43]; minimum points per cluster of ~4xalpha value were used for data analysis. For each condition, at least 5 ROIs were imaged and analyzed for each of the 3 independent replicates.

### 4.8. Statistical Analysis

Categorical variables were compared using Fisher’s exact test. Continuous variables were summarized as the mean ± standard error of the mean (S.E.M.) or average percentages ± S.E.M. Shapiro-Wilk test was used to measure data distribution normality of all continuous variables. The non-parametric Mann-Whitney U test was used for all non-normally distributed variables. All statistical tests were two-tailed, and *p* < 0.05 was considered statistically significant. Plots were conducted using Prism 10 (GraphPad, Boston, MA, USA). For SEVEN analyses, *p*-values for EV count were calculated with ANOVA using Prism 10 (GraphPad, Boston, MA, USA); log transformation was first applied to EV parameters that are not normally distributed (diameter, molecule count, and circularity), then *p*-values were then calculated with ANOVA on the logarithmic transformed data using Prism 10 (GraphPad, Boston, MA, USA); all data was plotted without logarithmic transformation.

## Figures and Tables

**Figure 1 ijms-25-11660-f001:**
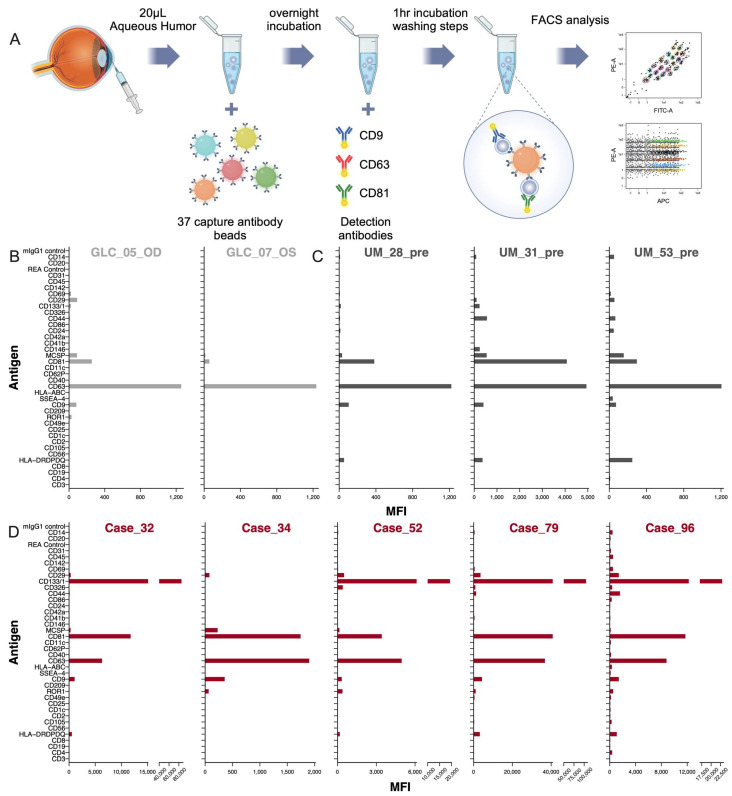
Surface marker profiles of EV/EPs in aqueous humor from RB, GLC, and UM cases were analyzed using multiplex bead-based flow cytometry. (**A**) Workflow design of the AH MACSPlex experiment. (**B**) Mean Fluorescence Intensity bar plots for two GLC aqueous humor samples, scaled to 5 μL. (**C**) Mean Fluorescence Intensity bar plots for three Uveal Melanoma samples collected prior to treatment. (**D**) Mean Fluorescence Intensity bar plots for five RB aqueous humor samples collected at the time of primary enucleation.

**Figure 2 ijms-25-11660-f002:**
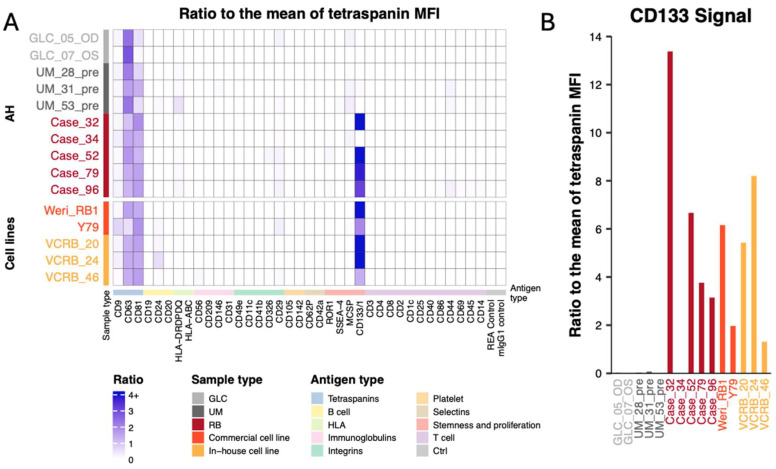
The ratio of surface marker mean fluorescence intensity (MFI) to mean tetraspanin MFI from RB, GLC, and UM cases using MACSPlex analysis. (**A**) Heatmap depicting the MFI for each surface marker compared to the mean CD9/63/81 MFI for each GLC AH sample, UM AH sample, RB AH sample, and RB cell line. (**B**) The ratio of CD133 tetraspanin signal to mean tetraspanin MFI for all AH samples and RB cell lines included in the analysis.

**Figure 3 ijms-25-11660-f003:**
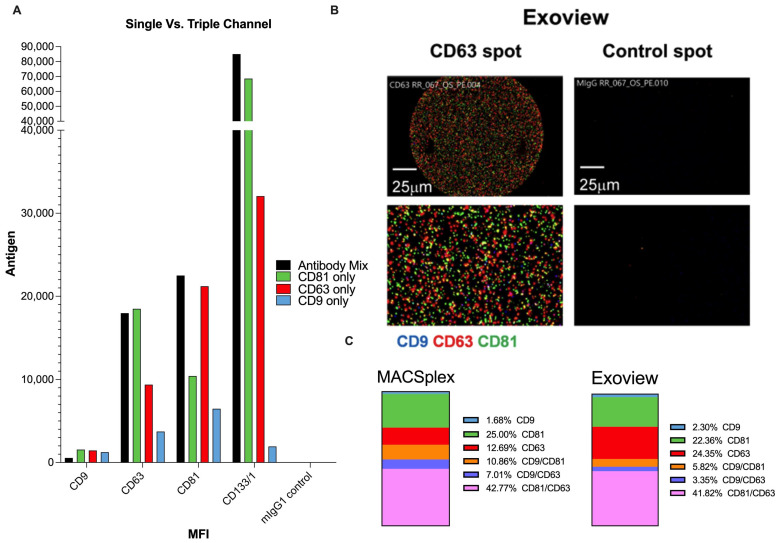
Concordance of MACSPlex Individual Channel Results and SP-IRIS Analysis in Aqueous Humor Sample Case 79. (**A**) Single and triple-channel MACSPlex experiments were conducted on the AH sample Case 79, where APC-conjugated CD9, CD63, or CD81 antibodies were individually added and processed through flow cytometry. Mean Fluorescence Intensity (MFI) analysis of CD9, CD63, CD81, and CD133 of an RB aqueous humor sample (Case 79) is presented for the antibody mix, CD9-only, CD63-only, and CD81-only experiments. (**B**) The same Case 79 AH underwent SP-IRIS analysis using the ExoviewR100 system for tetraspanin expression profiling. Representative fluorescent images were captured by fluorescent-conjugated antibodies (red = CD63-AF647, green = CD81-AF555, and blue = CD9-AF488). (**C**) Subpopulation breakdown of single and double positive EV populations using MACSPlex results compared to Exoview on Case 79.

**Figure 4 ijms-25-11660-f004:**
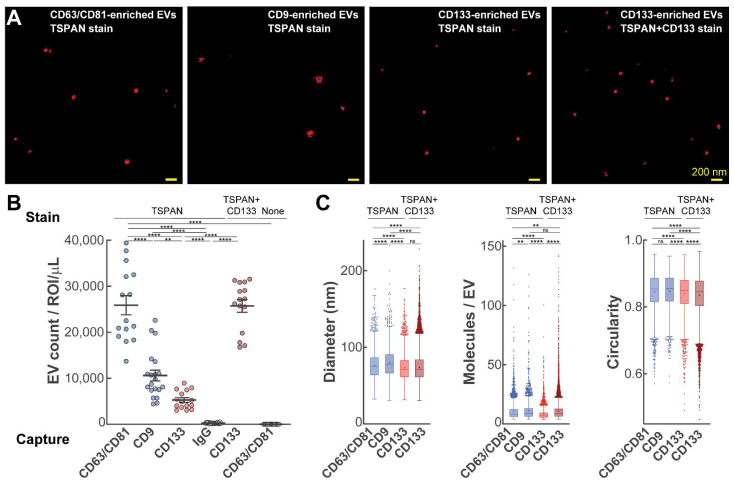
Multiparametric characterization of single EVs from AH using SEVENi. (**A**) Representative raw SMLM images of zoomed-in regions; sample dilutions are indicated in the methods section. (**B**) Number of detected EVs per region of interest normalized for 1 μL of AH; Error bars, SEM. (**C**) Distribution sizes, molecular contents, and circularity for detected EVs. Boxes denote interquartile ranges, center lines denote medians, crosses denote means, and the dots denote EVs beyond 1.5 times the interquartile range. *n* = 3 technical replicates (15 ROIs) for CD63/CD81 capture and CD133 capture, *n* = 4 technical replicates (20 ROIs) for CD9 capture and IgG capture; ** *p* < 0.01; **** *p* < 0.0001; ns: not significant. Values and statistics are included in Appendix A.

**Table 1 ijms-25-11660-t001:** Clinical demographics for samples included in the experiment. Clinical information was collected for all RB patients whose AH samples were included.

Study ID	Case 32	Case 34	Case 52	Case 79	Case 96
Gender	F	F	F	M	M
Age at dx (mon.)	10	16	15	64.6	24
Laterality	unilateral	bilateral	unilateral	unilateral	unilateral
Type of Mutation	none	RB1 Positive	none	none	none
IIRC Group	D	D	D	E	D
TNM classification	CT3D	CT2B	CT2B	CT3B	CT2B
IOP at dx	21	11	22	32	20
Type of seeding at diagnosis	Dust	none	sphere	sphere	cloud
Height at dx (mm)	9.6	9.93	10.9	>10	8.61
Base at dx (mm)	11.88	15.07	13.54	>10	17.7

## Data Availability

The data that support the findings of this study are available from the corresponding author upon request.

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
