# Peer review of "Phenotypic Biomarkers of Aqueous Extracellular Vesicles from Retinoblastoma Eyes"

_ijms, 2024, doi:10.3390/ijms252111660_

Round 1
Reviewer 1 Report
Comments and Suggestions for Authors
Thank you for inviting me to review this interesting manuscript.
With the growing focus on molecular biomarkers for non-invasive cancer diagnosis, this article objectively describes a potential biomarker in the aqueous humor of the eyes in retinoblastoma. I find it very thought-provoking.
Minor revisions suggested:
Abstract:
it is not clear which eyes formed the control group. Please clarify this.
Introduction:
Reference 19, 20: It would be interesting to discuss how sEVs play a role in tumor formation and progression. - CD 63/81+ EVs are abundant in pre-treatment RB eyes, particularly those with higher tumor burden. Is there a study that describes how much the sEVs of CD 63/81+ decrease after treatment?
Result:
Were AH uveal melanoma samples used as controls? Why did you use commercial cell lines for flow cytometry? Was it not possible to use cell lines derived from patients included in the study?
Discussion:
- Please discuss the study by Nail et al.: why do you think "the population of the CD133 cell surface signal within the Rb Y79 cell line was correlated with specific characteristics such as size, colony-forming ability, invasiveness potential, drug resistance, and gene expression pattern"?
- Why do you think "Rb Y79 EVs show a lower CD133 enrichment ratio compared to Weri RB EVs"? Please discuss further.
- Clarify the role of CD133 and CD81 in uveal melanoma.
Materials and Methods:
The p-values are not clearly described in the results and figures. Please correct this.
Comments on the Quality of English Language
No
Reviewer 2 Report
Comments and Suggestions for Authors
the submission by Amacker et al describes their pioneer work in retinoblastoma diagnosis from AH biopsy, characterising surface markers of EV by flow cytometry and SEVEN. current tests for retinoblastoma are limited to imaging, whilst biopsy creates a risk of damage and tumour spread.
The clear limitations of the current study are the sample size and the sensitivity.
I would suggest the authors to also cite the following and discuss:
10.1167/iovs.65.1.18
10.1136/bjophthalmol-2018-313005
